# New Earwigs from the Middle Jurassic Jiulongshan Formation of Northeastern China (Dermaptera) [note 1]

**DOI:** 10.3390/insects14070614

**Published:** 2023-07-07

**Authors:** Yuqing Yin, Chungkun Shih, Michael S. Engel, Dong Ren

**Affiliations:** 1College of Life Sciences, Capital Normal University, Beijing 100048, China; yinyuqing2023cnu@163.com (Y.Y.); chungkun.shih@gmail.com (C.S.); 2Department of Paleobiology, National Museum of Natural History, Smithsonian Institution, Washington, DC 20013-7012, USA; 3Division of Invertebrate Zoology, American Museum of Natural History, Central Park West at 79th Street, New York, NY 10024-5192, USA; mengel@amnh.org

**Keywords:** Protodiplatyidae, Semenoviolidae, earwigs, new genus, new species, systematic palaeontology

## Abstract

**Simple Summary:**

In this paper, two new genera with two new species of Dermaptera are described from the Middle Jurassic Jiulongshan Formation of Daohugou, Inner Mongolia, China. The discovery of these two new species enriches the comparatively meagre fossil record of Dermaptera, particularly from the Middle Jurassic. The description of *Applanatiforceps angustus* is another brick laid in the foundation of protodiplatyid diversity, and the new genus *Ekpagloderma gracilentum* highlights the diversity of cercal forms among Aglyptodermatinae and the remarkable homogeneity of the general morphology of groups within this clade.

**Abstract:**

Two new genera and species of Dermaptera are described from the Middle Jurassic Jiulongshan Formation of Daohugou, Inner Mongolia, China: *Applanatiforceps angustus* gen. et sp. nov. in the archidermapteran family Protodiplatyidae, and *Ekpagloderma gracilentum* gen et sp. nov. in the eodermapteran family Semenoviolidae. *Applanatiforceps* shares the typical characters of the extinct suborder Archidermaptera (e.g., pentamerous meta tarsi, filiform and multimerous cerci) and externalized ovipositor. The family identity of the Protodiplatyidae can be further distinguished by comparing this new genus with other genera of the Protodiplatyidae. As a result of its large compound eyes, tegmina without venation, body sparsely setose, legs rather short and slender, and shape of the veinless tegmina, *Ekpagloderma* is classified in the subfamily Aglyptodermatinae. *Ekpagloderma* not only has the typical features of the Aglyptodermatinae, but also exhibits a more primitive slender segmented cerci, which is different from all other genera of Eodermaptera. In fact, the diversity of Eodermaptera as known today indicates some of the challenges in understanding the suborder and whether or not it is monophyletic as historically construed, or if the separation of Turanodermaptera is justified.

## 1. Introduction

Dermaptera, commonly known as earwigs, are a small order of polyneopteran insects and includes about 2000 extant species [1]. Earwigs have hard leathery forewings, called tegmina, fan-shaped hind wings with a unique venation consisting of largely an expanded anal fan, and a pair of pincer-like cerci at the end of the abdomen [2,3]. As has been the case with many of the minor insect orders, the systematics of living and fossil Dermaptera has not received the intensity of investigation it deserves. Massalongo reported the discovery of a fossil of Forficulidae in the Eocene strata of Italy, and with this find opened the curtain on the study of fossil Dermaptera [4]. However, it was not until the 20th and first decades of the 21st century before fossil, and living, Dermaptera would be more extensively explored by a variety of scholars [5,6,7,8,9,10,11,12,13,14,15,16,17,18,19,20,21,22,23,24,25,26,27,28,29,30,31,32], with each reporting many complete fossil specimens from deposits all over the world and over a wide span of ages. Twenty years ago, Engel proposed a revised classificatory system for the order, which has been accepted by most researchers [16]. In particular, Engel noted that the traditional suborders of early authors (i.e., Archidermaptera, Hemimerina, Arixeniina, Forficulina) were artificial and obscured phylogenetic relationships among major lineages of earwigs, and he therefore reclassified the families into a new arrangement, with three suborders: Archidermaptera, Eodermaptera, and Neodermaptera [16,28].

Relative to other suborders, the Archidermaptera have many distinctive differences: pentamerous metatarsi, elongate, fine, multimerous cerci, a prominent externalized ovipositor, and often ocelli [16]. Engel recognized three archidermapteran families, Protodiplatyidae, Turanoviidae, and Dermapteridae [16,28], and with species found from the Late Triassic to Early Cretaceous [33]. Currently, there are 33 species of Archidermaptera in 20 genera and three families, and these have been documented from the Late Triassic, Early Jurassic, and Early Cretaceous of Britain; the Late Triassic of Australia; the Middle Jurassic in Inner Mongolia of China; the Upper Jurassic of Karatau, Kazakhstan; the Lower Cretaceous of Shandong Province, China; and the Lower Cretaceous Yixian Formation in Liaoning Province, China (Table 1).

Eodermaptera, comprising the extinct families of Semenoviolidae, Turanodermatidae, and Bellodermatidae (although turanodermatids have been classified as Turanodermaptera and sister to Neodermaptera relative to other eodermapterans: e.g., [17]), are distinguished from Archidermaptera by their trimerous pro- and mesotarsi, tegmina with simplified venation, and short, stubby cerci with either one or only a few cercomeres. The first eodermapteran discovered was *Semenoviola* obliquotruncata from Kazakhstan [6], which was followed by the later discovery of additional species from the same deposits. These additional species consisted of three species assigned to *Semenoviola* and one in a separate genus, *Turanoderma* [10]. Subsequently, Engel noted that the characters of *Turanoderma* were significantly different from semenoviolids and transferred the genus to the family Turanodermatidae [16]. Until now, only three families, five genera, and seven species of Eodermaptera have been reported from China and Kazakhstan.

Herein we describe two new genera and two new species of the families Protodiplatyidae (Archidermaptera) and Bellodermatidae (Eodermaptera) from the Middle Jurassic Jiulongshan Formation of Inner Mongolia, China. To date, a total of 32 species of Dermaptera from 10 families and 22 genera have been reported from China (including those described herein), spanning from the Middle Jurassic to the Miocene. This discovery not only increases the known diversity but also further expands our limited knowledge of the two extinct suborders Archidermaptera and Eodermaptera. The description of a new genus and a new species of Protodiplatyidae enriches our knowledge of Archidermaptera, highlighting the diversity of the group during the Jurassic as well as the observed morphological disparity across the clade.

The discovery of a new genus and species of Bellodermatidae similarly expands our knowledge of early diverging Eodermaptera, a group notable for their phylogenetic position [16,33,34]. Eodermaptera are more closely related to extant earwigs than to the more archaic Archidermaptera, and, in fact, Grimaldi and Engel united Eodermaptera and Neodermaptera into the clade Pandermaptera, as distinct from archidermapterans [33]. Many scholars have suggested that Eodermaptera are a transitional group between Archidermaptera and Neodermaptera, although such language that implies a linear progression of evolution is false and prevents a flawed depiction of the natural world and the processes that drive evolutionary change [35,36,37,38]. To avoid such misleading terminology, it should perhaps simply be noted that Eodermaptera as currently constituted is perhaps a grade and therefore obscures the proper sequence of divergence events. Phylogenetic analyses are few for fossil Dermaptera, but one such analysis recovered Bellodermatidae as the earliest-diverging Eodermaptera and that the Pandermaptera (=Eodermaptera + Neodermaptera) was monophyletic [33,34]. The same study indicated that Eodermaptera were paraphyletic relative to Neodermaptera, largely based on Turanoderma [34], the same taxon Engel had highlighted as needing removal to its own clade [39], Turanodermaptera (thereby removing some of the paraphyly). Note that owing to a failure of Zhao et al. to set their analytical program to collapse unsupported nodes, there are a number of seemingly “resolved” nodes in their tree which lack any character support and are therefore spurious and false (e.g., the placement of Semenovioloides, among others such as the “clades” within Protodiplatyidae or Dermapteridae) and so their assertion that Semenoviolidae are paraphyletic is not supported by their analysis and simply an error in the use of phylogenetic packages [34]. Thus, while the remaining groups of Eodermaptera may still be paraphyletic, there is no robust evidence for such an indication at this time. Accordingly, the best approximation of relationships within Pandermaptera as based on those nodes with actual robust character support in Zhao et al. can perhaps be summarized as (Table 2): Eodermaptera s.str. + (Turanodermaptera + Neodermaptera), where Eodermaptera s.l. of the past was Eodermaptera s.str. + Turanodermaptera, but this older, broader sense should be avoided for now and until more robust phylogenetic estimations are available.

## 2. Materials and Methods

Four specimens were collected from the Middle Jurassic Jiulongshan Formation at Daohugou Village, Ningcheng County, Inner Mongolia, northeastern China. The material is housed in the Key Laboratory of Insect Evolution and Environmental Changes, College of Life Sciences, Capital Normal University, Beijing (CNUB; Dong Ren, Curator). During the past 20 years, many species of well-preserved fossil insects, have been described from the Middle Jurassic Yanliao Entomofauna in Northeastern China, especially in Daohugou [40,41].

Detailed photographs were taken using a Nikon SMZ 25 microscope with a Nikon DS-Ri 2 digital camera (Nikon, Tokyo, Japan). Line drawings were prepared using the Adobe Illustrator CC and Adobe Photoshop CS5 (Adobe Systems, San Jose, CA, USA) software packages. Morphological terms used here follow those of Engel and Haas [28].

## 3. Results

Systematic palaeontology.

Order Dermaptera de Geer, 1773

Suborder Archidermaptera Bey-Bienko, 1936

Family Protodiplatyidae Martynov, 1925

Genus *Applanatiforceps*, Yin, Shih, Engel, and Ren gen. nov.

Type species. *Applanatiforceps angustus*, Yin, Shih, Engel, and Ren sp. nov.

Diagnosis. Moderate-sized earwigs, densely punctate-granulose throughout. Head slightly broad and flat, longer than wide, nearly rounded laterally; posterior border nearly straight, about as wide as the pronotum; surface lacking ecdysial cleavage scar; antenna with 21 antennomeres; robust and slightly broader scape than remaining antennomeres; pedicel short; all flagellomeres longer than wide; compound eyes small; ocelli absent. Pronotum approximately oval, anterior and posterior margins subequal in width. Tegmina with some longitudinal veins, anterior margin slightly curved, posterior margin somewhat truncate, outer margin arched and tapering posteriorly; squamata covering abdominal segment III. Femora not carinulate; all tarsi pentamerous (i.e., tarsal formula 5-5-5 rather than 4-4-5 in some genera); tarsi concave medially; pretarsal claws simple, arolium absent. Abdomen cylindrical, individual segments transverse, wider than long, anterior and posterior margins straight; cerci slender, with at least 28 cercomeres.

Etymology. The new genus-group name is a combination of the Latin *applānatus* (a combination of the prefix *ad*–, which becomes *ap*– before words beginning in –*p*–, and serves as an emphasizer; the noun *plānum*, meaning, “level” or “flat” ground; and the suffix –*ātus*, which is used to form adjectives from nouns indicating resemblance), meaning, “flattened”, and the noun *forceps*, meaning, “tongs” or “pincers”. The gender of the name is masculine.

Remarks. The new genus can be recognized as Protodiplatyidae based on the characteristically filiform antenna with 17–23 antennomeres; three simple veins reading tip of forewing (corresponding to Rs, M, and Cu); pentamerous metatarsus; cerci elongate, slender, and multimerous; and sclerosis of parameres in males. To date, Protodiplatyidae comprise a total of 11 genera including *Applanatiforceps*. Key diagnostic characters of the head size, number of antennomeres, tegmina, pronotum, and cerci for genera of Protodiplatyidae are summarized and compared in Table 3 to highlight their generic variations and similarities. Based on a combination of the following key differentiating characters, shown in Table 3, some of the key distinctions of *Applanatiforceps* from other genera are:The posterior margin of the head of *Applanatiforceps* is as wide as the pronotum (vs. posterior margin of the head of *Abrderma*, *Longicerciata*, and *Microdiplatys* broader than the pronotum; the posterior margin of the head of *Archidermapteron*, *Asiodiplatys*, *Perissoderma*, *Sinoprotodiplatys*, *Barbderma*, and *Protodiplatys* are narrower than the pronotum);*Applanatiforceps* has longitudinal venation in the tegmina (vs. *Aneuroderma*, *Sinoprotodiplatys*, *Asiodiplatys*, *Turanovia*, and *Longicerciata* without visible veins in the tegmina);The wings of *Applanatiforceps* and *Perissoderma* meeting the anterior margin of tergum III (vs. reaching the first, second, or fourth abdominal segments in other genera);The tarsal formula of *Applanatiforceps* is 5-5-5 (vs. 4-4-5 in *Archidermapteron*, *Asiodiplatys*, *Microdiplatys*, and *Protodiplatys*).

*Applanatiforceps angustus*, Yin, Shih, Engel, and Ren sp. nov.

Figure 1, Figure 2, Figure 3, Figure 4, Figure 5 and Figure 6.

urn:lsid:zoobank.org:act:EFC14B36-CA44-4FE8-A759-6D25FA1CF171.

Type material. Holotype, a completely preserved male, CNU-DER-NN2023002C/P (part and counterpart; Figure 1, Figure 2, Figure 3 and Figure 4); paratype, CNU-DER-NN2023003C/P (part and counterpart; Figure 5), and CNU-DER-NN2023004C/P(part and counterpart; Figure 6). All type material deposited in the College of Life Sciences, Capital Normal University, Beijing, China.

Etymology. The specific name is the Latin adjective *angustus*, meaning, “contracted” or “constricted”, and refers to the shape of the abdomen.

Locality and horizon. Jiulongshan Formation (Middle Jurassic); Daohugou Village, Wuhua Township, Ningcheng County, Inner Mongolia, China.

Diagnosis. As for the genus (*vide supra*).

Description. Adult male, preserved in both dorsal and ventral aspects. Body densely sculptured; without setae. Total length as preserved (excluding antennae and cerci) about 10.62 mm. Head medial length from clypeal apex to posterior border 1.17 mm, maximum width (across level of compound eyes) 1.12 mm, subquadrilateral (Figure 2E); posterior border straight; maxillary palpus pentamerous; antennal length 4.00 mm, with 21 elongate antennomeres; scape thick, subcylindrical, apex slightly expanded, broader than remaining antennomeres, length 0.42 mm, apical width 0.26 mm; pedicel shortest, length 0.10 mm; flagellomeres I and II equal in length, 0.15 mm; ocelli absent; compound eye small, length 0.72 mm.

Pronotum approximately circular and almost as broad as posterior margin of head; medial length 1.18 mm, maximum width 1.02 mm; anterior margin 0.86 mm wide, posterior margin 0.95 mm wide, both anterior and posterior margins slightly convex and lateral margins convexly rounded.

Tegmina present, length 2.52 mm, maximum width 0.82 mm, lateral margins arc-shaped, posterior margins truncate, squamata extending well beyond tegminal apex, truncate; Rs and M shorter and slightly straighter, almost parallel before Rs arches to terminate into M; Cu simple; tegmina and squamata covering abdominal terga I–II, and meets leading edge of abdominal tergum III.

Pro-, meso-, and metacoxae almost equidistant from each other; pro- and mesofemora cylindrical; femora not carinulate dorsally or ventrally; tibiae slightly more elongate and slightly flatter than corresponding femora; tarsi pentamerous; pretarsal claws present, simple, arolium absent. Abdomen cylindrical, long, narrow, almost all segments wider than long with apical margins straight, abdominal length as preserved (excluding cerci) 4.59 mm, maximum width 1.27 mm; cerci 2.42 mm long, longer than one-half abdominal length, with at least 28 elongate cercomeres; cercomere I longer than subsequent cercomeres; cercomeres about 0.43 mm wide, without setae.

Clade Pandermaptera Grimaldi and Engel, 2005

Suborder Eodermaptera Engel, 2003, s.str.

Family Semenoviolidae Vishniakova, 1980

Subfamily Aglyptodermatinae Xiong, Engel, and Ren, 2021

Genus *Ekpagloderma*, Yin, Shih, Engel, and Ren gen. nov.

Type species. *Ekpagloderma gracilentum*, Yin, Shih, Engel, and Ren sp. nov.

Diagnosis. Sparsely setose earwigs lacking distinct sculpturing (as in Aglyptodermatinae). Head broad, significantly broader than anterior border of pronotum, posterior margin nearly straight. Antenna with at least 12 antennomeres (incomplete as preserved; Aglyptodermatinae with 13 antennomeres); scape robust and slightly broader than remaining antennomeres; pedicel slightly longer than wide; all flagellomeres longer than wide. Compound eyes large and situated at posterior temples; ocelli absent (as in Aglyptodermatinae). Dorsal surface without Y-shaped ecdysial cleavage scar. Pronotum much broader than head, approximately oval, lateral margin slightly rounded, posterior margin centrally convex in a rounded shape (as in Aglyptodermatinae); length about 3× as wide, sides slightly curved and flared outward (perhaps explanate in life), posterior margin slightly rounded and inclined posteriorly. Legs with prominent setae; femora compressed and carinulate ventrally; tarsi trimerous, basitarsus elongate, tarsomere III articulating with apex of tarsomere II; pretarsal claw well developed, arolium absent. Female with exposed ovipositor. Cerci filiform, long, multimerous but much shorter than abdominal length, cercomere I more developed and elongate than remaining cercomeres, remaining cercomeres much longer than wide and gradually and progressively thinning to cercal apex.

Etymology. The new generic name is a combination of the Ancient Greek adjective *ἔκπαγλος* (*ékpaglos*, meaning, “marvelous” or “wonderous”) and the noun *δέρμᾰ* (*dérma*, meaning, “hide” or “skin”; genitive *δέρματος*/*dérmatos*). The gender of the name is neuter.

Remarks: The genus can be placed in Aglyptodermatinae based on the following characters: large compound eyes bordering posterior margin of head, Y-shaped ecdysial scar absent, posterior margin of head straight, pronotum broad and distinctly broader than head, tegmina without venation, and tegmina long and with outer margins diverging midlength or later before tapering as an arch to apex [41]. In addition, as with previously known Aglyptodermatinae the body is sparsely setose, legs are rather short and slender, the antennae are identical (cf. herein with [41]), the tegmina of both genera extend to abdominal segment III but do not meet the base of segment IV, the metatibiae are comparatively short, and the tarsi are trimerous. The new genus differs from Aglyptodermatinae, hitherto the sole genus of the subfamily, most notably as follows:The cerci of *Ekpagloderma* are not reduced to a single elongate cercomere, and are instead filiform, with cercomere I similar to that of Aglyptodermatinae, but followed by a series of thin, elongate cercomeres.Metatarsomere I (=metabasitarsomere) of *Ekpagloderma* is elongate and more than twice as long as metatarsomere II, rather than metatarsomere I short, as with metatarsomere II, in *Aglyptoderma*.*Ekpagloderma* retains the semenoviolid plesiomorphy of tarsomere II not extending slightly ventral the base of tarsomere III, while Aglyptodermatinae has an apomorphically augment character-state in which tarsomere III arises from the apical dorsal surface of tarsomere II [41].

It is tantalizing to see the cerci of *Ekpagloderma*, specifically the more developed and elongate first cercomere, as a character-state leading to the seemingly more apomorphic condition of Aglyptodermatinae, whereby all cercomeres beyond the first are lost [41], but the mode developed and elongate form of cercomere I is retained. In the absence of more taxa and a fully resolved phylogeny it is impossible to talk of specific changes between character states, but the limited data available do at least suggest an interesting hypothesis to be tested against future data. It should also be noted that Aglyptodermatinae are somewhat similar to the enigmatic protodermapteran subfamily Astreptolabidinae [41], a suite putative parallelism common to both lineages and which should be more fully explored as additional material of both groups is discovered in the future.

*Ekpagloderma gracilentum*, Yin, Shih, Engel, and Ren sp. nov.

Figure 7, Figure 8 and Figure 9.

urn:lsid:zoobank.org:act:F23FB07D-94CC-4233-9F75-26018FDDDF70.

Type material. Holotype. A completely preserved female, CNU-DER-NN2023001C/P, deposited in the College of Life Sciences, Capital Normal University, Beijing, China.

Etymology. The specific epithet is the Latin adjective *gracilentus*, meaning, “slender”, referring to the thin, elongate cerci.

Locality and horizon. Jiulongshan Formation (Middle Jurassic); Daohugou Village, Wuhua Township, Ningcheng County, Inner Mongolia, China.

Diagnosis. As for the genus (*vide supra*).

Description. Adult female, preserved in both dorsal and ventral aspects. Medium-sized. Total length as preserved (excluding antennae and cerci) about 8.04 mm, body with sparse pubescence. Head medial length about 2.08 mm (from clypeal apex to posterior border), maximum width (across compound eyes) 1.03 mm. Antennal length 1.97 mm, with at least 12 elongate antennomeres; scape thick, broader than remaining antennomeres, longer than wide, length 0.18 mm, apical width 0.15 mm; all flagellomeres longer than wide and longer than scape, flagellomere I longest of preserved flagellomeres. Compound eyes large and prominent, located near posterior margin of head, compound eye length 0.41 mm, interocular distance 0.55 mm; ocelli absent.

Pronotum approximately oval, lateral margin slightly rounded, posterior margin centrally convex and rounded, pronotum 0.91 mm long, 1.48 mm wide at widest point, 1.02 mm wide along leading edge, 1.32 mm wide along trailing edge. Tegmina well-developed, without venation, tegmina covering abdominal terga I and II, slightly overlapping tergum III.

Femora compressed carinulate ventrally; tibiae almost as long as femora, but more slender; tarsi trimerous; length of foreleg about 2.56 mm, profemur length 0.97 mm, protibia length 0.80 mm, length ratio of individual protarsomeres 0.22:0.16:0.21; length of mid-leg about 2.62 mm, mesofemur length 0.95 mm, mesotibia length 0.80 mm, length ratio of individual mesotarsomeres 0.30:0.16:0.25; length of hind-leg about 3.31 mm, metafemur length 1.10 mm, metatibia length 0.97 mm, length ratio of individual metatarsomeres 0.60:0.19:0.24; pretarsal claws present, simple, arolium absent.

Abdomen cylindrical, width tapering gradually toward apex; all segments wider than long, transverse, with comparatively straight anterior and posterior margins; abdominal length 4.27 mm (excluding cerci and valvulae), maximum width 1.70 mm.

## 4. Discussion

While there have been several taxonomic contributions on Protodiplatyidae, deeper exploration of their diversity has yet to be undertaken (e.g., morphometrics for studies of variation and disparity), and even the alpha taxonomy of the family needs further consideration (e.g., detailed redescriptions and imaging of those species described by Martynov and Vishniakova). Nonetheless, the discovery and documentation of new taxa adds to the available data for such future analyses, and enhances our understanding of the lineage in regards to biogeographic and temporal occurrences, morphological variety among species, and alpha diversity. In this context, the description of *Applanatiforceps angustus* is another brick laid in the foundation of protodiplatyid diversity. The genus is easily assigned to the family as it exhibits all of the usual characters for Protodiplatyidae, such as the reduced number of tegminal veins, and differs from previously known genera by its almost rounded pronotum, smaller compound eyes, and squamata extending to the anterior margin of the third abdominal segment. These are not dramatic departures from the usual suite of traits in the family, but certainly the combination of character-states is novel.

Perhaps of greater interest, however, is the discovery of *Ekpagloderma gracilentum*, an eodermapteran that, while generally similar to Aglyptodermatinae, notably expands our understanding of Aglyptodermatinae not only in alpha diversity but also in morphological variety regarding the cerci. As was mentioned for Protodiplatyidae, Eodermaptera are not well explored in depth, although some phylogenetic resolution has been attempted. Admittedly, there is less material of Eodermaptera and so this certainly hinders their exploration, to which the discovery of the new species certainly improves that situation, albeit in a minor fashion. Nonetheless, *Ekpagloderma* emphasizes some growing patterns that seem to emerge for these earwigs. Eodermaptera, as originally circumscribed (i.e., Eodermaptera s.l.), is certainly variable for a number of interesting traits. For example, if one takes three characters usually used in dermapteran systematics—ocelli, tegminal veins, and cercomere fusion—we find a growing and complex number of combinations and variations among Eodermaptera s.l. When originally proposed, the group included species of two small families (Semenoviolidae and Turanodermatidae) with trimerous tarsi and fused cercomeres (as with Neodermaptera) but retained tegminal veins and ocelli [16,28]. Later, the family Bellodermatidae was added, which, as with other Eodermaptera, retained tegminal veins, but lacked ocelli and had multimerous cerci [34]. When Aglyptodermatinae were later described, the number of combinations expanded further, with this lineage having a single cercomere (fusion or reduction?), combined with an absence of ocelli and tegminal veins (as with Neodermaptera) [41]. Thus, since 2010, Eodermaptera began to include lineages that muddied the concept of the group [34,41], with conflicting character combinations that suggested paraphyly relative to Neodermaptera, at least once these newer groups were added. Alternatively, there remained some characters continuing to support eodermapteran monophyly, specifically in females the constriction of terga VIII and IX, separate from tergum X, and not covered by tergum VII [41]. If the latter character continues to hold, then Turanodermaptera would simply fall as a synonym of Eodermaptera. If not, then what is the proper pattern of cladogenetic events? Discerning patterns is increasingly challenging for the group, particularly as *Ekpagloderma* demonstrates that Aglyptodermatinae originally had multimerous cerci and the apomorphic condition of Aglyptodermatinae is likely not homologous with that of Neodermaptera (the former seemingly the result of a loss of cercomeres apical to the basal cercomere, the latter seemingly the result of fusion of basal cercomeres combined with loss of all succeeding cercomeres). Discerning phylogenetic patterns and determine whether these hypotheses of character-state changes will hold shall require the discovery of additional Eodermaptera, emphasizing the need for further paleontological exploration [15], and ideally from further Jurassic and Early Cretaceous deposits so as to avoid the existing biogeographic bias in our knowledge of these early earwigs.

## 5. Conclusions

This work documents two new genera and two new species of Jurassic-aged earwigs: *Applanatiforceps angustus* gen. et sp. nov. of the Protodiplatyidae and *Ekpagloderma gracilentum* gen. et sp. nov. of the Semenoviolidae. All four of the newly reported specimens were collected from the Jiulongshan Formation of Daohugou in eastern Inner Mongolia, China. The discovery of these two new species enriches the comparatively meagre fossil record of Dermaptera, particularly from the Middle Jurassic. The new genus of protodiplatyids is generally similar to other taxa within the family, but nonetheless expands our limited understanding of the family’s biodiversity at the time as well as its overall morphological disparity. The genus, and its typical familial characters, accords with prior hypotheses regarding the pattern of evolution among early lineages of Dermaptera regarding the overall form of the cerci from elongate, flexible structures with large numbers of cercomeres to more rigid and shortened forms in later clades. The other genus, *Ekpagloderma*, expands notably out understanding of the eodermapteran subfamily Aglyptodermatinae, a subfamily previously known from two species in a single genus from the same deposits. The genus highlights the diversity of cercal forms among Aglyptodermatinae, and the remarkable homogeneity of the general morphology of groups within this clade. Collectively, these new taxa are an enticing progression in our understanding of Mesozoic Dermaptera, and before they assumed the characteristic forcipate cerci that are so diagnostic of modern earwigs.

## Figures and Tables

**Figure 1 insects-14-00614-f001:**
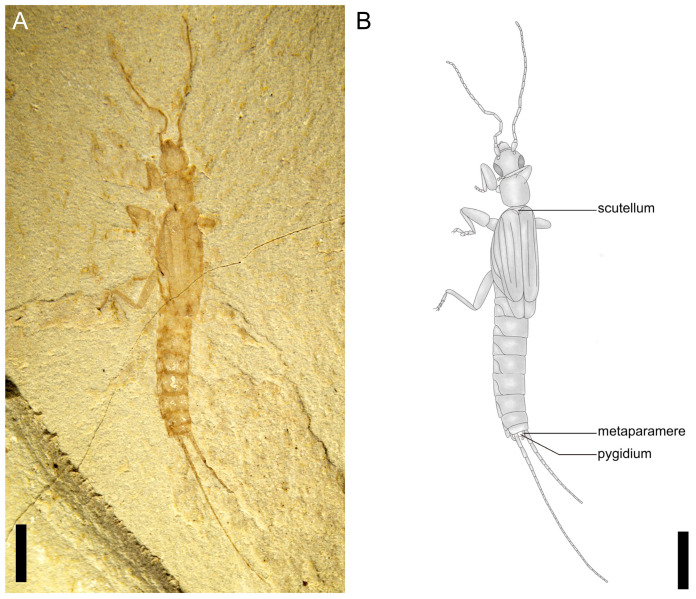
*Applanatiforceps angustus* gen. et sp. nov., Holotype (CNU-DER-NN2023002C), male. (**A**) Photograph of dorsal aspect. (**B**) Line drawing of dorsal aspect. Scale bars: 2.0 mm (**A**,**B**).

**Figure 2 insects-14-00614-f002:**
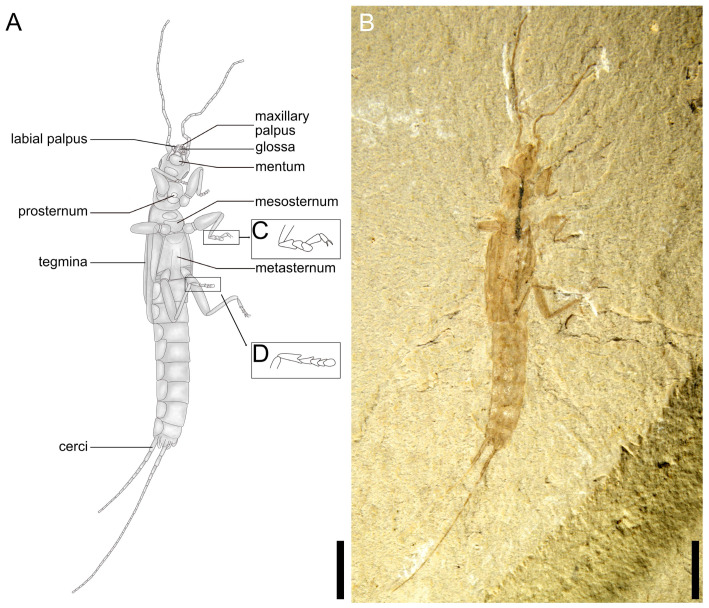
*Applanatiforceps angustus* gen. et sp. nov. Holotype (CNU-DER-NN2023002P), male. (**A**), Line drawing of ventral aspect. (**B**) Photograph of ventral aspect. (**C**) Anterior lateral (prolateral) view of left mesotarsus. (**D**) Anterior ventral (proventral) view of left metatarsus. Scale bars: 2.0 mm (**A**,**B**). The animal is slightly twisted as preserved, with the head and anterior thoracic segments in ventral position, while the metathorax and abdomen are obliquely lateral.

**Figure 3 insects-14-00614-f003:**
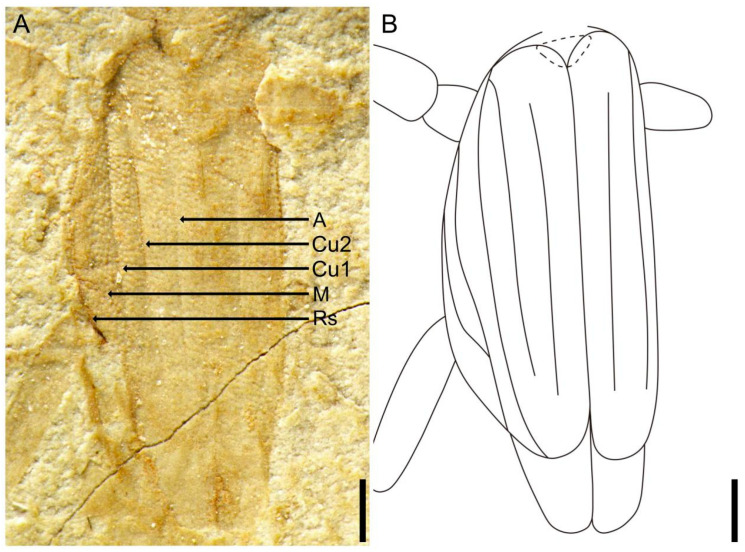
*Applanatiforceps angustus* gen. et sp. nov. Holotype (CNU-DER-NN2023002P). (**A**) Photograph of tegmina in dorsal aspect. (**B**) Line drawing of tegmina in dorsal aspect. Scale bars: 0.5 mm (**A**,**B**).

**Figure 4 insects-14-00614-f004:**
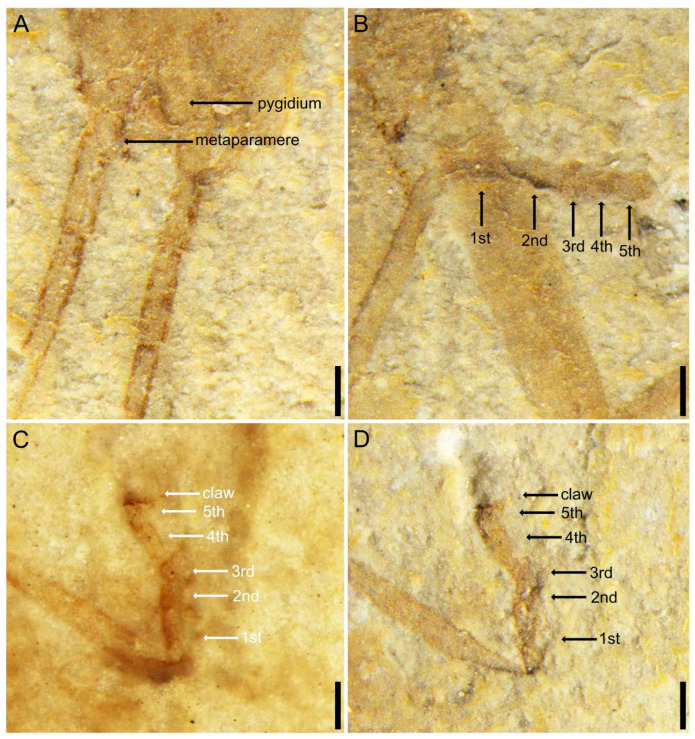
*Applanatiforceps angustus* gen. et sp. nov. Holotype (CNU-DER-NN2023002C), male. (**A**) Ventral aspect of terminal abdominal segment and basal cercomeres. (**B**) Anterior ventral aspect of metatarsus. (**C**) Anterior lateral (prolateral) view of mesotarsus, viewed submerged in ethanol. (**D**) Anterior lateral (prolateral) view of mesotarsus, viewed dry. Scale bars: 0.2 mm (**A**–**D**).

**Figure 5 insects-14-00614-f005:**
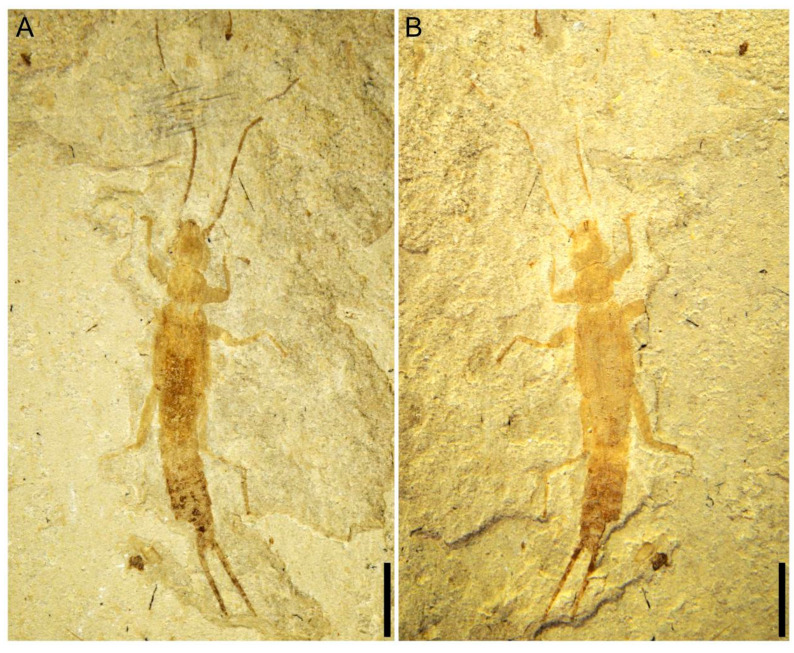
*Applanatiforceps angustus* gen. et sp. nov. Paratype (CNU-DER-NN2023003C/P), male. (**A**) Photograph of dorsal aspect. (**B**) Photograph of ventral aspect. Scale bars: 2.0 mm (**A**,**B**).

**Figure 6 insects-14-00614-f006:**
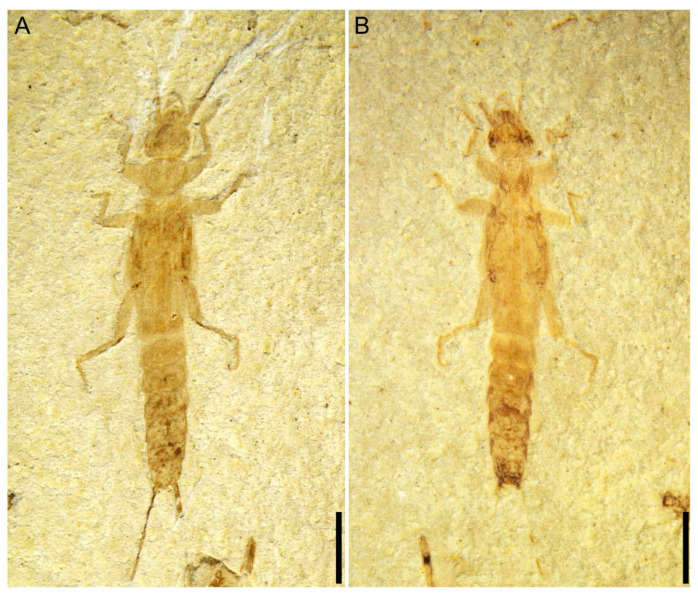
*Applanatiforceps angustus* gen. et sp. nov. Paratype (CNU-DER-NN2023004C/P). (**A**) Photograph of dorsal aspect. (**B**) Photograph of ventral aspect. Scale bars: 2.0 mm (**A**,**B**).

**Figure 7 insects-14-00614-f007:**
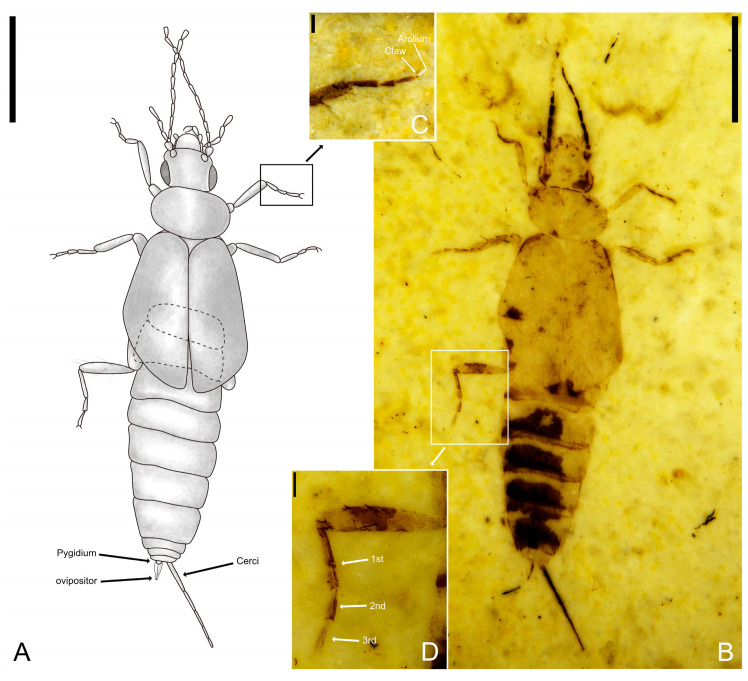
Holotype of *Ekpagloderma gracilentum* gen. et sp. nov. CNU-DER-NN2023001C. (**A**) Line drawing of dorsal aspect. (**B**) Photograph of dorsal aspect, viewed submerged in ethanol. (**C**) Anterior lateral (prolateral) view of right protibial apex and protarsus. (**D**) Anterior lateral (prolateral) view of left metatibia and metatarsus. Scale bars: 2.0 mm (**A**,**B**); 0.2 mm (**C**,**D**).

**Figure 8 insects-14-00614-f008:**
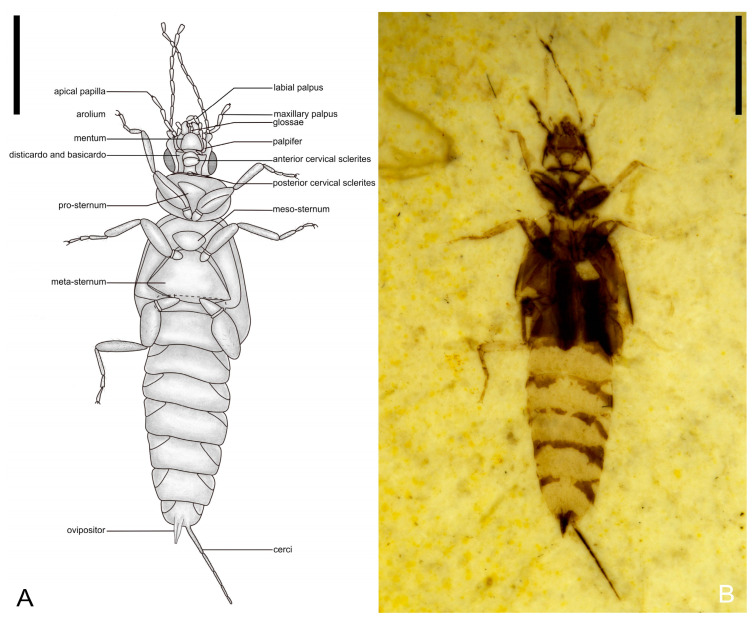
Holotype of *Ekpagloderma gracilentum* gen. et sp. nov. CNU-DER-NN2023001P. (**A**) Line drawing of ventral aspect. (**B**) Photograph of ventral aspect, viewed submerged in ethanol. Scale bars: 2.0 mm (**A**,**B**).

**Figure 9 insects-14-00614-f009:**
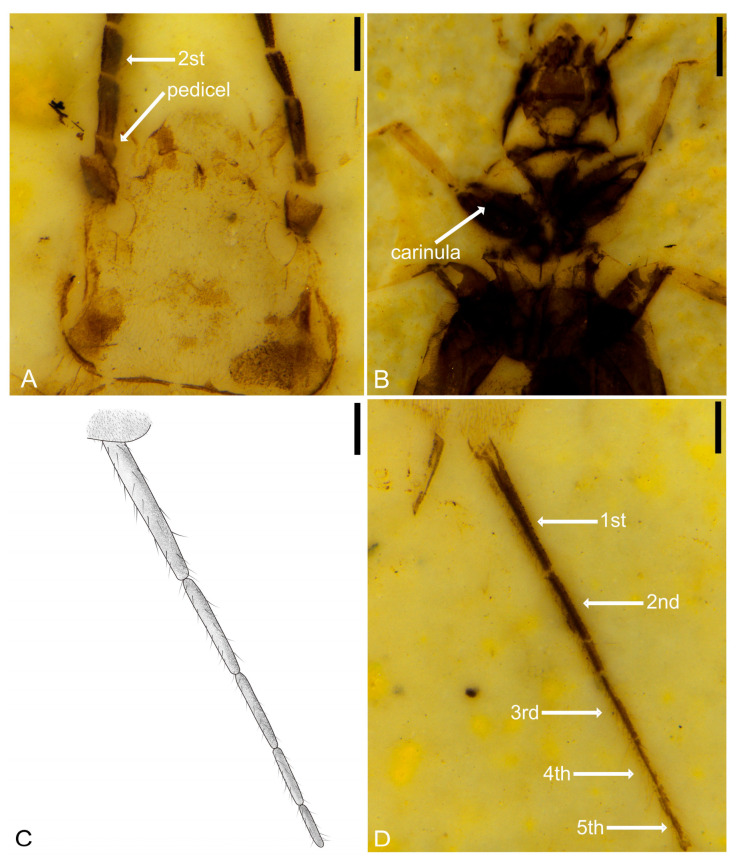
Holotype of *Ekpagloderma gracilentum* gen. et sp. nov. CNU-DER-NN2023001P, viewed submerged in ethanol. (**A**) Dorsal view of head. (**B**) Ventral view of head. (**C**) Line drawing of cercus. (**D**) Dorsal aspect of cercus. Scale bars: 0.5 mm (**B**); 0.2 mm (**A**,**C**,**D**).

**Table 1 insects-14-00614-t001:** Checklist of described Archidermaptera.

Family	Genus	Species	Locality	Epoch
Protodiplatyidae	*Abrderma*	*Barbderma oblonguata* Xing, Shih, and Ren, 2016	China	J2
	*Aneuroderma*	*Aneuroderma oiodes* Xiong, Engel, and Ren, 2021	China	J2
	*Archidermapteron*	*Archidermapteron martynovi* Vishnyakova, 1985	Kazakhstan	J3
	*Asiodiplatys*	*Asiodiplatys speciosus* Vishnyakova, 1985	Kazakhstan	J3
	*Barbderma*	*Barbderma oblonguata* Xing, Shih, and Ren, 2016	China	K1
	*Longicerciata*	*Longicerciata mesozoica* Zhang, 1994	China	J2
		*Longicerciata rumpens* Zhang, 1994	China	J2
	*Microdiplatys*	*Microdiplatys campodeiformis* Vishnyakova, 1985	Kazakhstan	J3
		*Microdiplatys oculatus* Vishnyakova, 1985	Kazakhstan	J3
		*Microdiplatys perfectus* Vishnyakova, 1985	Kazakhstan	J2
	*Perissoderma*	*Perissoderma triangulum* Xing, Shih, and Ren, 2016	China	J2
	*Protodiplatys*	*Protodiplatys fortis* Martynov, 1925	Kazakhstan	J3
		*Protodiplatys gracilis* Vishnyakova, 1980	Kazakhstan	J3
		*Protodiplatys mongoliensis* Vishnyakova, 1986	Mongolia	K1
	*Sinoprotodiplatys*	*Sinoprotodiplatys ellipsoideuata* Xing, Shih, and Ren, 2016	China	K1
		*Sinoprotodiplatys zhangi* Nel, Aria, and Garrouste, 2012	China	K1
Dermapteridae	*Brevicula*	*Brevicula gradus* Whalley, 1985	United Kingdom	J1
		*Brevicula maculata* Kelly, Ross and Jarzembowski, 2018	United Kingdom	J1
	*Dacryoderma*	*Dacryoderma teres* (Tihelka, 2019)	United Kingdom	J1
	*Dermapteron*	*Dermapteron incerta* Martynov, 1925	Kazakhstan	J
	*Dimapteron*	*Dimapteron corami* Kelly, Ross, and Jarzembowski, 2018	United Kingdom	K1
		*Jurassimedeola orientalis* Zhang, 2002	China	J2
	*Palaeodermapteron*	*Palaeodermapteron dicranum* Zhao, Shih, and Ren, 2011	China	J2
		*Phanerogramma australis* Kelly, Ross, and Jarzembowski, 2018	Australia	T3
		*Phanerogramma dunstani* Kelly, Ross, and Jarzembowski, 2018	Australia	T3
		*Phanerogramma gouldsbroughi* Kelly, Ross, and Jarzembowski, 2018	United Kingdom	J1
		*Phanerogramma heeri* Giebel, 1856	United Kingdom	J1
		*Phanerogramma kellyi* Kelly, Ross, and Jarzembowski, 2018	United Kingdom	T3
	*Sinopalaeodermata*	*Sinopalaeodermata neimonggolensis* Zhang, 2002	China	J2
		*Sinopalaeodermata concavum* Xiong, Engel, and Ren, 2021	China	J2
	*Trivenapteron*	*Trivenapteron moorei* Kelly, Ross, and Jarzembowski, 2018	United Kingdom	J1
	*Valdopteron*	*Valdopteron woodi* Kelly, Ross, and Jarzembowski, 2018	United Kingdom	K1
Turanoviidae	*Turanovia*	*Turanovia incompleta* Vishniakova, 1985	Kazakhstan	J3

**Table 2 insects-14-00614-t002:** Hierarchical arrangement of major lineages of Dermaptera, with daggers (†) indicating extinct groups. Ranks are not included for the time pending a comprehensive review of the clades. Families for Archidermaptera and Eodermaptera are included for reference as these lineages are addressed herein. Groups marked with an asterisk (*) could be paraphyletic and require more extensive sampling of taxa and character data in future cladistic analyses. An alternative possibility is that Apodermaptera should be composed of Eodermaptera s.str. and Neodermaptera, with Turanodermaptera sister to this clade, but there is the possibility that Eodermaptera s.l. is actually monophyletic (figure 4 therein, [40]).

† Archidermaptera Bey-Bienko († Protodiplatyidae, † Dermapteridae, † Turanoviidae)
Pandermaptera Grimaldi and Engel
† Eodermaptera Engel, s.str. († Bellodermatidae, † Semenoviolidae)
Apodermaptera Engel
† Turanodermaptera Engel († Turanodermatidae)
Neodermaptera Engel
Nematodermaptera Engel * (Karschielloidea, Diplatyoidea, Haplodiplatyidae)
Gamodermaptera Engel
Protodermaptera Zacher, s.str. * (Pygidicranidae *)
Epidermaptera Engel
Paradermaptera Verhoeff *
Metadermaptera Engel *
Eteodermaptera Engel
Plesiodermaptera Engel
Eudermaptera Verhoeff

**Table 3 insects-14-00614-t003:** Particular characters of genera of Archidermaptera known from more than just isolated tegmina. Interrogative marks (?) indicate data unavailable.

Genus	Head	Antennomere	Tegmina	Pronotum	Tarsal Formula
*Abrderma*	Broader than pronotum	17–19	With venation, overlapping segment I, but not reaching anterior margin of segment II	Elliptical, anterior and posterior margins subequal in width	?-?-5
*Aneuroderma*	As wide as pronotum	20	Without venation, covering abdominal segment II	Oval, anterior and posterior margins subequal in width	5-5-5
*Archidermapteron*	Narrower than pronotum	17–19	With venation, covering abdominal segment IV	Reniform, broad notch anteriorly	4-4-5
*Asiodiplatys*	Narrower than pronotum	22	Without venation, covering abdominal segment II	With shallow, broad notch anteriorly	4-4-5
*Barbderma*	Broader than pronotum	19	With venation, overlapping anterior of abdominal segment I	Oblong or trapezoidal, anterior and posterior margins subequal in width	?-?-?
*Longicerciata*	Broader than pronotum	more than 26	Without venation, reaching anterior of abdominal segment IV	Transverse, anterior margin wider than posterior margin	5-5-5
*Microdiplatys*	Broader than pronotum	19	With venation, reaching anterior of abdominal segment IV	Transverse	4-4-5
*Perissoderma*	Narrower than pronotum	17	With venation, slightly beyond anterior margin of abdominal segment III	Elliptical, anterior margin wider than posterior margin	5-5-5
*Protodiplatys*	Narrower than pronotum	17–18	With venation, reaching anterior of abdominal segment IV	Transverse, notch in front, broadly rounded posteriorly	4-4-5
*Sinoprotodiplatys*	Narrower than pronotum	18	Without venation, reaching anterior of abdominal segment IV	Anterior and posterior margins subequal in width	5-5-5
*Applanatiforceps gen. nov.*	As wide as pronotum	21	With venation, reaching anterior margin of abdominal segment III	Approximate circle, anterior and posterior margins subequal in width	5-5-5

## Data Availability

All data from this study are available in this paper and the associated papers.

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
