# Peer review of "New Earwigs from the Middle Jurassic Jiulongshan Formation of Northeastern China (Dermaptera)†"

_insects, 2023, doi:10.3390/insects14070614_

Round 1

Reviewer 1 Report

This is an important contribution based on exceptionally preserved specimens of the fascinating order Dermaptera. Apart of the indications above, the figures are excellent. For the first time, I have nothing substantial to criticize in respect to the scientific content of a manuscript reviewed by me. Congratulations to the authors. I only have to indicate very minor corrections:

 lines 90-126...: please, the genera and species in italics

line 95: no comma before Turanodermatidae

line 107: "similar" maybe better "similarly"

line 238 and line 239: "Figures" changes to "Figure"

line 275: "nov" changes to "nov."

Figures 1 and 2: labelling too much small and difficult to read

line 434: "Scale bars,"   changes to "Scale bars:"

lines 438 and 439: please note that the authors of taxa sometimes using ", &" and others " and". Please use a unique format along the manuscript

line 453: "well-developed" maybe better as "well developed"

Lastly, in my opinion the "5. Conclusions" section contains some results mixed with conclusions and in "4. Discussion" section seems contains some conclusions. I read these two sections and everything is clear and well presented, but maybe the authors could consider these two sections to maybe grouping all the conclusions in the last section.

Author Response

-This is an important contribution based on exceptionally preserved specimens of the fascinating order Dermaptera. Apart of the indications above, the figures are excellent. For the first time, I have nothing substantial to criticize in respect to the scientific content of a manuscript reviewed by me. Congratulations to the authors. I only have to indicate very minor corrections:

 lines 90-126...: please, the genera and species in italics

line 95: no comma before Turanodermatidae

line 107: "similar" maybe better "similarly"

line 238 and line 239: "Figures" changes to "Figure"

line 275: "nov" changes to "nov."

Figures 1 and 2: labelling too much small and difficult to read

line 434: "Scale bars,"   changes to "Scale bars:"

lines 438 and 439: please note that the authors of taxa sometimes using ", &" and others " and". Please use a unique format along the manuscript

line 453: "well-developed" maybe better as "well developed"

Reply: Thank you for your kind suggestion. I have diligently addressed all the identified issues, including formatting, image captions, word choice, and punctuation usage.

-Lastly, in my opinion the "5. Conclusions" section contains some results mixed with conclusions and in "4. Discussion" section seems contains some conclusions. I read these two sections and everything is clear and well presented, but maybe the authors could consider these two sections to maybe grouping all the conclusions in the last section.

Reply: Thank you for your suggestion. The section of the conclusion is a brief summary of the content of the article, and we think it is more convenient for the reader to understand the content of the whole article by listing it separately, so we have kept the two sections of discussion and conclusion.

Reviewer 2 Report

Reading such excellent and solid  contribution to the systematics and biodiversity of Dermaptera was a pleasure. Information on the various subfamilies is comprehensive and instructive. Additions were clearly made to the incorrect prior classifications.

Line drawings and plate figures are appropriate and instructive.

Lines: 90, 92, 93 check: Semenioviola, Turanoderma

Author Response

-Reading such excellent and solid  contribution to the systematics and biodiversity of Dermaptera was a pleasure. Information on the various subfamilies is comprehensive and instructive. Additions were clearly made to the incorrect prior classifications.

Line drawings and plate figures are appropriate and instructive.

Lines: 90, 92, 93 check: Semenioviola, Turanoderma

Reply: Thanks for your kind suggestion. I have rectified the formatting issues mentioned earlier.